# Silicon versus Superbug: Assessing Machine Learning’s Role in the Fight against Antimicrobial Resistance

**DOI:** 10.3390/antibiotics12111604

**Published:** 2023-11-08

**Authors:** Tallon Coxe, Rajeev K. Azad

**Affiliations:** 1Department of Biological Sciences, University of North Texas, Denton, TX 76203, USA; talloncoxe@my.unt.edu; 2BioDiscovery Institute, University of North Texas, Denton, TX 76203, USA

**Keywords:** antibiotic, antimicrobial resistance, machine learning, antibiotic discovery

## Abstract

In his 1945 Nobel Prize acceptance speech, Sir Alexander Fleming warned of antimicrobial resistance (AMR) if the necessary precautions were not taken diligently. As the growing threat of AMR continues to loom over humanity, we must look forward to alternative diagnostic tools and preventive measures to thwart looming economic collapse and untold mortality worldwide. The integration of machine learning (ML) methodologies within the framework of such tools/pipelines presents a promising avenue, offering unprecedented insights into the underlying mechanisms of resistance and enabling the development of more targeted and effective treatments. This paper explores the applications of ML in predicting and understanding AMR, highlighting its potential in revolutionizing healthcare practices. From the utilization of supervised-learning approaches to analyze genetic signatures of antibiotic resistance to the development of tools and databases, such as the Comprehensive Antibiotic Resistance Database (CARD), ML is actively shaping the future of AMR research. However, the successful implementation of ML in this domain is not without challenges. The dependence on high-quality data, the risk of overfitting, model selection, and potential bias in training data are issues that must be systematically addressed. Despite these challenges, the synergy between ML and biomedical research shows great promise in combating the growing menace of antibiotic resistance.

## 1. Understanding the Scale of AMR

In 1928, Sir Alexander Fleming inadvertently ushered in the modern era of antibiotics, following his groundbreaking discovery of penicillin. With the onset of the Second World War, penicillin would become mass-produced and prove to be instrumental in the treatment of wounded soldiers, given its impressive chemotherapeutic properties. This significantly reduced the mortality rate that was previously caused by common infectious diseases, marking a revolutionary milestone for global healthcare. After garnering worldwide popularity as a wonder drug, penicillin became widely available to the public. This spurred searches for other antibiotics, culminating in the identification of multiple antibiotic classes. Unfortunately, the decades following these medical breakthroughs have seen deleterious consequences to public health. This is mainly attributed to two reasons: first, the rampant misuse of antibiotics—particularly in developing countries [1]—has led to the proliferation of numerous antibiotic-resistant bacterial strains. Second, the golden age of antibiotic discovery dramatically dwindled after the 1960s, which led to a “discovery void” (Figure 1); wherein very few new antibiotics have been brought to light, further compounding the issue [2].

As of 2023, antimicrobial resistance (AMR) is among the greatest threats to human health, with an estimated 1.27 million deaths being directly attributed to AMR infections in 2019 [4] and a projection that such deaths will reach ~10 million per year by 2050 [5]. AMR arises from bacterial pathogens’ coevolved resistance mechanisms, rendering the use of antibiotics ineffective or drastically reducing the efficacy of treatment. This means that previously antibiotic-susceptible bacteria may confer resistance to the antibiotics, following acquisition of such traits via mechanisms such as mutation or horizontal gene transfer (HGT). Pathogenic bacterial strains may gain resistance determinants that give rise to superbugs—often those of yet unknown genotypic composition. This poses, potentially, the greatest risk to vulnerable healthcare patients, as hospitals are hotspots for HGT, resulting in life-threatening infections with extremely limited treatment options [6]. In certain instances, resistant strains of *Escherichia coli* and *Klebsiella pneumoniae* have resulted in infections that are impervious to all known antibiotics, including carbapenems, a class of drugs that is typically reserved as the last line of defense in treating bacterial infections [7]. One notable example was an incident that occurred in 2016, when an infected patient in Nevada developed septic shock and died from a *Klebsiella pneumoniae* strain that was resistant to all 26 antibiotic classes available in the United States and, therefore, was given the informal title of the “nightmare superbug” [8]. The coexistence of susceptible microorganisms with resistant bacteria facilitates the exchange of AMR genes, especially within a spatial population, such as a biofilm [9].

The emergence of superbugs among bacteria is closely intertwined with an evolutionary phenomenon of gene exchange among different lineages—namely, the horizontal gene transfer (HGT) or the lateral gene transfer (LGT)—which is driven by three primary mechanisms: conjugation, transduction, and transformation. These mechanisms facilitate the transfer of resistance genes between bacteria, albeit through distinct processes. Conjugation involves the direct transfer of plasmids, which may carry resistance genes, between bacterial cells. Transduction entails the transfer of DNA via bacteriophages. Transformation involves bacteria taking up free DNA fragments from their environments. While each of these processes facilitates the lateral transfer of foreign DNA into a host bacterial cell, each mechanism differs vastly and may signify a genomic signature that is specific to the respective mechanism [10]. 

Understanding the nuances of HGT mechanisms is vital in the context of antibiotic resistance, as these processes drive the spread of resistance genes across bacterial populations, exacerbating the challenge of treating bacterial infections. Thus, it is imperative that necessary precautions be undertaken to combat antimicrobial resistance. The overuse and misuse of antibiotics, poor infection control practices, and inadequate investment in research and development are major players in promoting the rise of AMR [11]. Figure 2 provides a visual representation of antibiotic prescription trends in the United States over the decade from 2011 to 2021, showcasing the five leading antibiotic classes and agents prescribed during that period [12]. Promoting the responsible use of antibiotics should be encouraged and the public should be made aware of the consequences of their misuse. Further monitoring of the spread of resistant bacteria should be conducted through the implementation of antimicrobial stewardship programs to reduce the dissemination of resistance genes and the impact of antibiotics on the environment and, further, by advocating for practices that safeguard both human well-being and ecological health.

Antibiotic usage in agriculture is responsible for a significant portion of the occurrence of resistant bacterial strains. The mass administration of antibiotics seeks to promote growth and hinder the prophylaxis of diseases among livestock [13]; it accounts for ~80% of total antibiotic usage in the United States [14]. It is estimated that by 2030, worldwide usage of antibiotics for livestock will be in excess of 107,500 tonnes; for comparison, this was under 100,000 tonnes in 2020 [15]. These practices, evidently, have led to the development of antibiotic-resistant bacteria in animals, which can then be transmitted to humans through the food chain, through animal–human interactions, or via environmental contamination. For instance, resistant bacteria are transmitted to humans from livestock via direct contact or exposure to animal manure. The consumption of undercooked meat and direct contact also contribute to the spread of resistant bacteria. Various environmental sources, such as soils, freshwater, and wastewater systems, can become reservoirs of AMR, partly due to runoff from agricultural practices [16]. When such environments are contaminated with antibiotic residues and resistant bacteria, they act as a conduit for the spread of AMR, impacting wildlife, pets, and humans [17]. Furthermore, these environmental resistomes—the collection of all antibiotic resistance genes in a microbiome—pose a substantial threat to public health. As bacteria can share resistance genes through horizontal gene transfer, even non-pathogenic environmental bacteria can contribute to the spread of AMR by acting as a reservoir of resistance genes [18]. Therefore, the unregulated and excessive use of antibiotics in agriculture not only enriches the pool of resistance genes but also contributes significantly to the global dissemination of antibiotic resistant bacteria. Regarding the global trends, it has been estimated that antimicrobial use in chicken, cattle, and pigs—accounting for >90% of food animals—was in excess of 93,000 tonnes in 2017 and is projected to reach 104,000 tonnes by 2030 [14]. Very recently, the United States Food and Drug Administration (FDA) has taken actions to combat AMR by supporting antimicrobial stewardship in veterinary matters. With effect from June 2023, the FDA requires animal owners to have a veterinary prescription to purchase antibiotics that were previously available over-the-counter and misused [19].

In the context of the escalating problem of AMR, there exists an urgent imperative for the discovery of new antibiotics. However, the pace of development of new antimicrobial drugs has markedly decelerated. During what is often referred to as the golden age of antibiotic discovery, research was equally focused on both naturally derived and synthetic antibiotics. However, following almost half a century of intensive screening, the pursuit of antibiotics from natural sources appears to have reached a saturation point [20]. Identifying and validating new bacterial targets, overcoming bacterial defense mechanisms, and ensuring that an antibiotic can reach its target site all contribute to the difficulty of the task [21]. Furthermore, the scientific challenges are compounded by economic and regulatory issues. Developing a new drug is costly and time-consuming, often requiring more than a decade and hundreds of millions of dollars to bring a drug from discovery to market [22]. However, new antibiotics may not be profitable because they are often kept as a last resort to prevent the development of resistance, which limits their use and, thus, their market potential. 

In light of these challenges, there is a need for innovative strategies and policies to stimulate antibiotic discovery. Regulatory hurdles can further delay or discourage the development of new antibiotics. Designing clinical trials for antibiotics is a complex process, requiring the comparison of a new drug to an existing standard-of-care treatment. This comparison is inherently challenging, due to the varying and sometimes unpredictable responses of patients with bacterial infections. Ultimately, the development of new antibiotics requires human efficacy data which, in turn, requires further interpretation in the context of other strong and relevant information that supports the effectiveness of the new therapy [23].

Over the course of the previous decades, several methods have been developed to assess an antibiotic’s ability to inhibit bacterial infection. Commonly used in modern clinical laboratories is in vitro susceptibility testing, also known as antibiotic susceptibility testing (AST). This method involves the exposure of isolated bacterial colonies to different antimicrobial agents. The minimum inhibitory concentration (MIC) is then measured via the inhibition zones on an agar plate to determine the sensitivity of each tested antibiotic [24]. AST is relatively cost-effective and easy to perform, providing valuable information on an antibiotic’s activity against any specific bacterial isolate; however, it is conjointly limited by its simplicity. In vitro AST does not account for the complexities of its host’s infection site or for host immune response. While MIC testing may aid in the observation of antibiotic resistance patterns, the results may not always correlate directly with its in vivo efficacy. This may be due to a variety of factors, including but not limited to an antibiotic’s limited ability to penetrate certain tissues, biofilm formation, host metabolism, and immune response. Comparable methods that also bear these limitations include disk diffusion, broth dilution, and E-test strips, all of which provide quantitative MIC values.

Biofilm formation, linked with chronic infections that result from indwelling devices, wounds, and cystic fibrosis lung conditions, is a frequent occurrence in hospital settings. Biofilms are the congregations of bacterial communities that can cling to both living and non-living surfaces. Moreover, these biofilms typically become entrenched in an extracellular matrix, an environment that fosters their interactions with host molecules [25]. Despite the fact that biofilm formation represents the standard mode of bacterial growth, clinicians predominantly employ planktonic inoculums when conducting almost all tests for antimicrobial resistance in pathogenic bacteria [26]. The word planktonic refers to free-living bacteria, i.e., those bacteria which are regarded as standard in traditional AST. Infective endocarditis (IE) is a condition characterized by the entrance of biofilm bacteria into the bloodstream, which then reside within cardiac tissue. 

Traditional diagnostic methods evaluate the blood culture of an IE-infected host; however, the results may return as a false-negative, since biofilm bacteria seldom enter the bloodstream in planktonic form [27]. Thus, immunodiagnostic assays have been employed to identify antibodies that are aimed at components of the biofilm matrix. One such instance includes ELISA, an immunodiagnostic assay that is specifically designed to detect antibodies that target slime polysaccharide antigens in staphylococci. However, the sensitivity and specificity of currently available ELISA assays are not sufficient to independently confirm the presence of biofilm-associated infections [28]. On the other hand, sessile bacteria, or those that lack mobility, display significantly increased AMR compared with their planktonic counterparts. Consequently, biofilm-derived pathogenesis contributes to the diminishing treatment options available to patients with chronic, resistant bacterial infections, and in some instances, biofilms may even contribute to cancer morbidity [29]. 

As previously stated, bacterial communities in a biofilm interact through their extracellular matrix with host molecules; however, biofilms may also form within the living cells of the host. This can cause collateral tissue damage by simultaneously potentiating both the innate and acquired immune response in the host. Especially in cases of chronic infections, the host remains in a perpetual inflammatory phase that is characterized by oxidative damage, fibroblast senescence, and a lack of beneficial growth factors that are needed for tissue resolution [30]. When biofilm pathogenesis and AMR coexist, they can significantly amplify host immune challenges. This situation is further complicated when existing antibiotic treatments are ineffective.

Conventional AST methods, used for their precision and clinical applicability, come with drawbacks, such as substantial time investment and relative costliness. In critical-care scenarios, these inherent delays might force physicians to opt for broad-spectrum antibiotics that are designed to combat a wide array of infections. However, these antibiotics, despite their “broad-spectrum” designation, may not always serve as the most effective treatment modality [31]. Even more concerning, their utilization has been scientifically linked to the growing phenomenon of AMR. Conventional cell culturing methods, with their limitations in mimicking the in vivo microenvironment of cells, call for the need to adapt two-dimensional techniques to a three-dimensional problem. This requires innovative methodologies and more comprehensive insights. In parallel, in the realm of drug discovery, the traditional means of determining antibiotic efficacy have spurred demand for quicker testing. Although new techniques are being developed to accelerate susceptibility testing, the time required for validation is often less than optimal, underscoring the necessity for further refinement and advancement [32]. 

Vocat et al. have spearheaded recent advancements in antibiotic susceptibility testing (AST) by devising a method to classify *Mycobacterium tuberculosis* (MTB) strains based on their resistance to isoniazid and rifampicin. Their innovative technique fuses nanomotion technology with machine learning to provide a swift and precise test for MTB susceptibility [33]. Unlike traditional MIC growth-dependent methods, which typically take 3–5 days for resistance profiling [34], this novel approach directly assesses bacterial reactions to antibiotics through nanomotion, reducing the turnaround time to less than 24 h. Further analysis of the raw nanomotion data (indicative of bacterial responses to antibiotics) entails inputting this information into a machine-learning model for training and testing. Their model had been implemented to classify bacteria using their nanomotion data into either susceptible or resistant groups, showing effective differentiation capabilities. This approach underscores the trend in the emergence of new computational techniques to augment the power of AST, which is indicative of a larger trend toward the integration of advanced technologies in microbiological research.

## 2. Machine Learning: An Overview

In today’s rapidly advancing technological landscape, we are witnessing an almost Renaissance-like resurgence in the growth and development of artificial intelligence (AI) and machine-learning (ML) models. AI is set to become an increasingly integral aspect of our everyday lives, as illustrated by the public’s awareness of OpenAI’s ChatGPT, the fastest-growing consumer application in history, with over 100 million active users. We have entered AI’s most accessible era yet, now allowing laymen to interface with powerful computational tools without any preliminary understanding of the underlying technology. AI has captured the attention of global CEOs as well, with 80% of Fortune 500 companies purportedly having integrated ChatGPT into their organization at the time of this writing [35]. Leveraging the capabilities of next-generation AI and machine-learning models, we can explore openly available disease-related datasets. This enables us to uncover hidden patterns, particularly functionally critical regions within multiomic sequencing data and beyond.

ML is concerned with fitting models, or mathematical representations of processes using algorithms, in order to elucidate relational patterns within data [36]. Consequently, we may leverage this technology to gain unprecedented insights into the intricate biological systems that underlie a wide range of diseases. This insight is gained through predictions or decisions made by the model, which can be further investigated. Generally speaking, ML is classified into two broad categories: supervised learning and unsupervised learning. In supervised learning, an ML model is trained using labeled data, which have already been characterized prior to training. Predictions and decisions made by a supervised model are akin to learning with a guiding hand. In contrast, unsupervised models are used to discover intrinsic patterns within data in order to gain insights surrounding the data. This lends itself to clustering, or abstract data grouping, based on the similarities or differences found within the dataset. In either case, these techniques greatly rely on the availability of high-quality data for the model to make accurate predictions and decisions.

Deep learning (DL), a subset of machine learning, is rooted in a mathematical framework known as the artificial neural network (ANN). Similar to the way cells are the basic units of life, nodes or neurons are the basic units of an ANN. ANNs are foundational computational models that are analogous to the functional aspects of biological neural networks in the human brain; they consist of interconnected nodes that process input data and yield certain outputs. These nodes are arranged in layers, with each layer executing a distinct computation. The output from one layer serves as the input for the subsequent layer, continuing until the final layer, which offers the prediction. Figure 3 provides a visual representation of this process; i.e.; a simplified depiction of a neural network, illustrating the flow of data from input to output, and the learning process involved. At the core of each neuron in the ANN is a fundamental equation that combines an input vector, X, with a weight vector, W, then adds a bias term, b, followed by the application of an activation function, *f*(). This relationship is mathematically represented as follows:$$a=f\left(\overset{L}{\underset{l=1}{\sum }}w_{l}x_{l-1}+b_{l}\right)$$
where *x*_1_, *x*_2,_…,*x_n_* are input features, *w*_1_, *w*_2_,...*w_n_* are their corresponding weights, *b* is the bias term, *f*() is the activation function, and *a* represents the neuron’s output.

Data are represented as input X, and corresponding labels as Y. In the process of training a machine-learning model, it is essential to meticulously divide the data into training, testing, and validation sets. This division serves distinct purposes: initiating model training, evaluating the model’s performance, refining the parameters to improve the model’s performance, and, finally, testing the model on a held-out test dataset. This is a foundational step in both classification and regression problems. Within this context, features—i.e., the known properties or characteristics of a given training dataset—serve as the input to the model, aiding in the identification of relevant information within a dataset. Within the context of bacterial studies, these features could include the genotypes (genes) and phenotypes (e.g., gram-negative or gram-positive, morphology, and mode of respiration) of bacteria. Note that the labels function as the output variables, representing what the model aims to predict (e.g., antibiotic-susceptible or antibiotic-resistant). Moreover, the labels used as output variables might also denote distinct bacterial resistance mechanisms—for example, assigning label 0 for “antibiotic target alteration” and label 1 for “antibiotic efflux”. In the realm of supervised learning, these labels are supplied with the training data for the model to learn the association between data features and the output variables. Unsupervised learning, however, forgoes labels, seeking patterns or groupings without preconceived outcomes.

To gauge the match of a model’s predictions with the actual target values, a loss function is deployed. This function quantifies the disparity between the predicted values and the true values. The procedure for minimizing this loss during the training phase aims to learn parameters that result in alignment of the model’s predictions with the corresponding known values. In turn, this bolsters its performance and accuracy. An optimization algorithm iteratively refines the process by updating the model’s parameters and computing the loss function to carry out the updating, based on the level of convergence between the predicted values and the true values. The challenge, here, is to strike a balance to avoid overfitting or underfitting the model to the training data. Overfitting occurs when a model fails to generalize from the training data, becoming ineffectual with new data, while an underfitting model misses the underlying data trends, yielding suboptimal performance. Ensuring a model’s robustness requires assessing its performance against yet-unseen data. Depending on the problem at hand, various metrics can be used for this assessment. For classification tasks, where the output is a discrete label, metrics such as accuracy, precision, recall, and *F*1 score are commonly used. For regression problems, where the output is continuous, metrics such as the mean squared error (*MSE*) are more suitable. Precision (*PR*) is calculated as follows:$$Precision=\frac{true\;positive}{true\;positive+false\;positive}$$

Sensitivity (*SN*) is calculated as follows:$$Sensitivity=\frac{true\;positive}{true\;positive+false\;negative}$$

The harmonic mean (*F*1) of precision and sensitivity is calculated as follows:$$F1\;score=2\times \frac{PR\times SN}{PR+SN}$$

The mean squared error is calculated as follows:$$MSE=\frac{1}{n}\sum_{i=1}^{n}{\left(Y_{i}-{\hat{Y}}_{i}\right)}^{2},$$
where *n* is the number of data points, *Y_i_* represents observed values, and *Ŷ_i_* represents predicted values.

In the final stages of model optimization, once satisfactory performance is reached, attention is often turned to the fine-tuning of hyperparameters. These are distinct from the parameters that the model learns during training. While parameters adapt, based on the data, hyperparameters are preset configurations that are integral to the model’s architecture. Challenges often arise in the successful implementation of ML; a significant challenge is choosing the best hyperparameters for the model to learn effectively. These hyperparameters might include choices related to the learning rate, the structure of hidden layers in a neural network, or other higher-level structural settings that guide the training process. The learning rate is a measure that determines the extent of changes made to the parameters during training. Choosing a suitable learning rate is crucial. If it is too high, the training process might miss the ideal configuration that minimizes the loss, hindering the convergence. On the other hand, if it is too low, the training may take an excessively long time to converge. The correct configuration of hyperparameters is crucial, as they govern fundamental aspects of model training, such as convergence speed and the model’s capacity to generalize from the training data. Through calibration, changes may be made to the model to enhance its learning ability and its decision-making ability—ultimately adapting the architecture to best suit the data and the task at hand.

## 3. The Application of Machine Learning to Biomedical Research

Within the rapidly evolving paradigm of AI and machine learning, a niche has been carved out that marries computational power with human-like decision-making. This interplay between data and algorithms transcends mere technological curiosity; their tangible impacts on human health are within our grasp. As we delve into the intricate world of genomics, proteomics, and complex biological systems, ML becomes more than a tool—it emerges as a partner in unraveling the mysteries of life and disease. With the open availability of massive datasets, nuanced algorithms, and precise hyperparameter tuning, we have forged a pathway toward understanding, predicting, and potentially reversing maladies that have plagued humanity. From predictive modeling in personalized medicine to real-time diagnostics in critical-care settings, machine learning is poised to revolutionize the way we approach, manage, and, ultimately, conquer disease.

ML’s application in computer science and technology is just the tip of the iceberg. Its integration with biomedical data has led to the development of predictive algorithms that can identify disease markers, analyze genetic mutations, and even assist in personalized medicine. Machine learning is frequently used in cancer diagnosis and detection, particularly in cancer prediction and prognosis [37]. It has been applied to improve the accuracy of predicting cancer susceptibility, recurrence, and mortality by 15–25% [38]. The integration of machine learning, particularly deep learning (DL), into digital image analysis has enabled rapid advances in computational pathology. Applications of ML methods in pathology have significantly improved the detection of metastases in lymph nodes [39], Ki67 scoring in breast cancer [40], Gleason grading in prostate cancer [41], and more. DL models have also been demonstrated to predict the status of molecular markers in various cancers [42]. 

AI and the Internet of Medical Things (IoMT) are being combined to design efficient point-of-care biomedical systems that are suitable for next-generation intelligent healthcare. AI’s role in supporting advanced robotic surgeries and improving the functionality, detection accuracy, and decision-making ability of IoMT devices has been discussed in recent research [43]. AI algorithms have also been developed for diagnosing and treating colorectal cancer [44], the third most diagnosed malignancy. AI-assisted techniques in routine screening represent a pivotal step in the decline in colorectal cancer morbidity. ML models have contributed to individual-based cancer care, including robotic surgery and computer-assisted drug delivery techniques [45]. Undoubtedly, the convergence of machine learning with biomedical applications is actively shaping the future of healthcare. By leveraging the computational power of modern computers and the analytical capabilities of ML, researchers and clinicians are unlocking new opportunities for understanding and combating diseases. This synergy heralds a new era of innovation and discovery in healthcare, demonstrating the versatility and potential of AI.

## 4. Machine Learning and Antimicrobial Resistance

Predicting AMR using ML methodologies often utilizes a supervised learning approach. In this approach, a dataset with known labels indicating antibiotic susceptibility and resistance phenotypes is used for training. The model learns to elucidate patterns in the features (e.g., gene sequences or MIC concentrations) to accurately predict these labels for bacterial pathogens [46].

ML’s applicability in AMR prediction tasks has seen its usage in various forms. ML has been applied to characterize antibiotic-resistant strains of *Escherichia coli* by utilizing a pan-genome approach to identify core gene clusters and antibiotic resistance genes (ARGs) [47]. That study demonstrated how ML can be used to annotate genomic FASTA files and predict resistance to specific antibiotics, achieving better prediction accuracy for AMR genes within the accessory part of the pan-genome and, thus, elucidating clusters of AMR genes that are not present in all strains of *E. coli.* Their approach demonstrated the uneven distribution of genes, particularly ARGs, which reside in the accessory pan-genome. Through the implementation of a genetic algorithm (GA) within their model, the prediction of ARGs and their corresponding clusters was performed by learning the presence/absence patterns of *E. coli* gene clusters downloaded from the Pathosystems Resource Integration Center (PATRIC) database. This GA discriminates between gene clusters through binary representations—annotating them as either including [1] or not including [30] certain genes in predicting AMR activities. A fitness function, based on the area-under-the-curve (AUC), estimated fitness for the support vector machine (SVM) training method. The GA process was repeated 30,000 times to determine all subsets of gene clusters and to establish those that are associated with AMR phenotypes; in turn, better prediction of resistance profiles could be analyzed for downstream analysis. Their analysis of the *E. coli* pan-genome combined re-annotation, clustering, machine learning, and genetic algorithms to identify key factors and to offer insights into the complexities of AMR.

Further, an SVM ensemble approach, trained using the pan-genomes of *Staphylococcus aureus*, *Pseudomonas aeruginosa,* and *Escherichia coli*, was developed to establish resistance determinants and to predict AMR phenotypes [48]. Similar to the model developed by Her and Wu, a binary labeling was used, which resulted in a sparse binary matrix. The pan-genomes of the three pathogens were encoded by the model, based upon the presence or absence of each gene and allele. By encoding these features into a binary matrix, the genomic variation between strains could then be sampled to train the SVM. Given a genotype, the model predicted the AMR phenotype by outputting binary values for an antibiotic of interest. The weights—the assigned values to represent the “strength” of the relationship between the features—were then averaged across the ensemble of models to rank features by their association to AMR. A detailed analysis of fluoroquinolone’s resistance profile revealed perfect segregation by the presence or absence of known AMR-conferring mutations. Their study identified 25 candidate AMR-conferring genetic features, several of which are of interest in potentially identifying resistance determinants at the gene level.

The two studies mentioned above utilized the PATRIC database, which is an excellent resource for genome-scale data for bacteria, particularly those that are pathogenic to humans [49]. However, there exist multiple databases dedicated to hosting specific types of AMR data. For example, APD3, DBAASP v3, dbAMP, and PhytAMP are all databases that are dedicated to antimicrobial peptides (AMPs). In recent years, the discovery of novel AMPs has garnered interest to address the growing challenge of AMR. As traditional antibiotics are becoming less effective, the unique mechanisms of actions of AMPs have been increasingly recognized for their capacity to inhibit pathogens from developing resistance against them. Due to the natural versatility and specificity exhibited by peptides, researchers are exploring ways to engineer AMPs for therapeutic application. Studying AMPs reveals a promising pathway toward developing novel drugs that may be able to be used concurrently with existing antibiotics, or even as standalone treatments [50]. 

AMPTrans-lstm is an approach that was created to produce and design diverse, novel functional peptides by employing a deep generative model [51]. This model combines the stability of long short-term memory (LSTM) with the novel application of transformer architecture. The integrated model includes a pre-trained phase on a large dataset, followed by fine-tuning on a smaller dataset. The model generates sequences through LSTM sampling and then decodes them into novel peptide sequences using the transformer model. In total, 36,000 AMP candidates were generated from AMPTrans-lstm and evaluated with support vector machine (SVM) and random forest (RF), then further examined using quantitative-structure-activity-relationship (QSAR) modeling to estimate the probability that the generated sequences would have antimicrobial properties. The success rate of AMPTrans-lstm is calculated to be between 30% and 50%, marking a step toward generating novel AMPs, of which only a few have historically advanced to clinical trials [52].

AMPlify is an attentive DL model that has identified 75 putative AMPs derived from the genome of *Lithobates catesbeianus* [53]. The model was trained and tested using data sourced from the Antimicrobial Peptide Database (APD3) [54] and the Database of Anuran Defense Peptides (DADP) [55], which contained 3061 and 1923 AMP sequences, respectively. After removing duplicates, a negative dataset of 4173 sequences was curated from the UniProtKB/Swiss-Prot database [56]. Both the positive and negative datasets were split into 80% for training and 20% for testing. The MAKER2 gene prediction pipeline [57] was employed to refine the sequences that aligned to the bullfrog draft genome, including two filtering stages: the selection of sequences characterized by a distinct lysine-arginine motif and a threshold of 200 amino acids. From these filtered sequences, AMPlify output a probability score, with a threshold of >0.5 indicating AMPs and ≤0.5 indicating non-AMPs. In all, AMPlify predicted 75 putative AMPs. Of those sequences, 11 were ultimately selected for in vitro testing, and four novel AMPs demonstrated significant potency as an antimicrobial when tested on the clinical multi-drug resistant (MDR) isolate of CPO *E. coli*. The bacterial isolates employed for this testing encompassed a spectrum, inclusive of strains resistant to multiple drugs. The outcomes of these tests were quantified in terms of MIC and minimum bactericidal concentrations (MBC). While the MIC values represented the peptide concentration threshold that inhibits visible bacterial proliferation, the MBC values signified the concentration required to exterminate 99.9% of the initial bacterial population. An MBC assay was performed only in select clinical scenarios, typically when a patient’s immune system was unable to effectively combat the pathogen—as observed in cases of endocarditis, osteomyelitis, and immunosuppressed patients who were diagnosed with neutropenia [58].

AMPlify’s architecture uses attention mechanisms during sequence analysis, assigning weights to each of the positions within a given sequence. After residues are one-hot encoded for preprocessing, they are passed through the three hidden layers, beginning with a bidirectional long short-term memory (Bi-LSTM) layer. From this first layer, positional information is encoded in a recurrent manner. A multi-head scaled dot-product attention (MHSDPA) layer follows to represent the sequence, using multiple weight vectors. The final hidden layer of context attention generates a single summary vector from a weighted average, leveraging contextual information learned earlier during training. Binary cross-entropy is employed as the model’s loss function, along with Adam optimization to adjust the weights. To counteract any overfitting, dropout is utilized—a regularization technique that “drops” or nullifies noisy activations in the model’s layers.

The HMD-ARG-DB database is the largest and most comprehensive ARG database to date. It is constructed by merging sequences from seven existing databases: CARD, ARDB, ResFinder, ARG-ANNOT, MEGARes, SARG, and NDARO [59]. It includes over 17,000 manually curated ARG sequences, allowing for ML models to capture the most relevant features associated with resistance phenotypes. By performing multi-level annotation by integrating data from various sources, HMD-ARG-DB bridges gaps between different ARG databases, offering a unified and standardized resource. Traditional computational methods for identifying ARGs are primarily based on sequence alignment, which are limited in their ability to identify new ARGs, due to reliance on prior characterization. To overcome these challenges, Li et al. introduced a novel hierarchical multi-task deep learning framework for ARG annotation (HMD-ARG). This ML framework can identify multiple ARG properties simultaneously; it can determine if a given protein sequence is encoded by an ARG, the antibiotic family it resists, its resistance mechanism, and whether it is intrinsic or acquired. Furthermore, if the predicted antibiotic family is beta-lactamase, HMD-ARG can also predict its subclass. To improve HMD-ARG’s ability to generalize, 66,000 non-ARG sequences from UniProt were added as negative examples during training. HMD-ARG’s architecture is based on an end-to-end convolutional neural network (CNN), which utilizes one-hot encoding for the input. These inputs come as strings of protein sequences composed of 23 characters that correspond to the different amino acids. After the sequences are one-hot encoded, representing them as vectors for the model to process, the data proceed through six convolutional layers and four pooling layers to learn statistical patterns and motifs within the input sequences. The outputs are piped into three fully connected layers to discern functional mapping patterns; each layer corresponds to the task of predicting the drug, mechanism, or source. This multitasking framework forces each connected layer to simultaneously discover features with a single forward-propagation. HMD-ARG performed with an F1-score of 0.948 when performing binary classification to discern between ARGs and non-ARGs. While surpassed by DeepARG’s [60] F1 of 0.963, HMD-ARG still outperformed CARD, DeepARG, AMRPlusPlus [61], and Meta-MARC [62] in classifying antibiotic classes and predicting antibiotic mechanisms. The overall robust performance of HMD-ARG makes it a valuable contribution to the field; however, its applications are limited when working with short reads. This means that inputting AMPs to HMD-ARG would likely yield an unfavorable performance, due to small peptide lengths compared to the greater lengths of gene sequences.

Identifying the components that characterize those bacterial strains that confer resistance to antibiotics becomes difficult without an understanding of the known genetic markers that are responsible for the resistance phenotype. Without this knowledge, the challenge intensifies, especially in machine-learning development. Selecting an appropriate classification algorithm to predict novel genetic features that contribute to antibiotic resistance adds an additional layer of complexity. Since no single optimal ML algorithm exists to predict resistance phenotypes across all bacterial species, conducting performance assessment through hyperparameter tuning and cross-validation becomes a necessary task. A framework for selecting the best performing model(s) that predict the most relevant AMR loci involved in resistance has been developed specifically for the purpose of predicting the phenotype, along with the identification of genetic factors that underlie resistance traits [63]. Sourcing all data from the isolates browser from the NCBI Pathogen Detection website, filtered by AMR and AST phenotypes, Sunuwar and Azad created a binary representation of all bacterial samples, based on genotype (0 for absence and 1 for presence of an ARG) and the relevant antibiotic phenotype (0 for AST, 1 for AMR). 

Following this representation of the data, a three-fold performance assessment was carried out. This includes three separate workflow instances, as follows.: all performance: metrics were derived from the entire AMR dataset, with a focus on cataloging AMR genes of high importance; intersection performance: metrics were based on genes that consistently ranked among the top AMR genes throughout the 6-fold cross-validation in the first workflow—specifically, “consistent” genes that were chosen from the top 30 high-importance genes in each cross-validation fold. These were identified as the most critical features that the machine-learning algorithm used to predict the susceptible and resistance phenotypes; 3. random sampling: metrics were calculated using randomly selected AMR genes, termed “random features”—the final performance measure was an average derived from 10 such random sets sampled from the entire dataset. They assessed the performance of 12 different ML algorithms using genotypic data from *K. pneumoniae*, *E. coli* and *Shigella*, *P. aeruginosa*, *C. jejuni* and *S. enterica* genotypic, and the respective phenotypic data sourced from tests with several antibiotics. These 12 algorithms—logistic regression (logR), Gaussian naive Bayes (gNB), support vector machine (SVM), decision trees (DT), random forest (RF), k-nearest neighbors (KNN), linear discriminant analysis (LDA), multinomial naive Bayes (mNB), AdaBoost classifier (ABC), gradient boosting classifier (GBC), extra trees classifier (ETC), and bagging classifier (BC)—were deployed using the scikit-learn library in Python.

Continuing to leverage ML toward addressing AMR, Sunuwar and Azad built upon their previous research using homology modeling and molecular docking to predict potential interactions of novel ARG products to different antibiotics. Using bacterial isolates filtered by genotypic and AST phenotypic data from the NCBI Pathogen Detection database, the researchers constructed an AMR-AST matrix for each combination of antibiotic-species groupings. This matrix consisted of the features (genes), binary AST labels as target classes, and sample accession numbers to be input into a variety of ML algorithms to perform supervised binary classification [64]. These algorithms were trained and tested using 6-fold stratified cross-validation—implemented in StratifiedKFold—for all genes and AST phenotype data to ensure that genes deemed important for discrimination were obtained. Recall, Precision, F1, AU ROC, and AUPR were used as performance metrics computed for both training and test datasets, with the highest overall accuracy (F1) on test datasets informing which model would be selected as the optimal model. 

As compared to the previous study of Sunuwar and Azad, which focused solely on AMR genes, this more recent study used all genes within the strains to identify genes which ones had yet to be implicated in resistance phenotypes, showcasing an unbiased whole-genome approach. The top-ranking putative novel AMR genes underwent homology modeling and molecular docking analyses. This led to the discovery of several modifying enzymes bearing catalytic activity functionally, similar to acetylation, phosphorylation, and adenylation. The researchers used AutoDock Vina v1.1.2 for docking [65], preparing the best protein data bank (PDB) models for receptors by removing water molecules and other heteroatoms, repairing hydrogens, and adding charges. The structured data file of the respective antibiotics (ligands) was sourced from PubChem and converted to PDB format. The receptors were then docked with the respective ligands, and ligand-receptor binding free energy was scored. Upon closer analysis, these enzymes were associated with steric hindrance, which decreases the affinity of antimicrobials and provides insight into their unique mechanisms of resistance. Ultimately, the proteins encoded by the novel ARGs displayed high binding affinity with their respective antibiotics in silico. This integrated approach of combining ML, homology modeling, and simulated molecular docking facilitated both the classification of novel AMR genes and the validation of their potential interactions with antibiotics within a unified framework. 

It should be emphasized that their unbiased whole-genome approach need not be limited to pathogens and may be extended to commensal microorganisms that act as reservoirs for pathogenic strains to gain resistance determinants via HGT. Harnessing the plethora of computational tools and resources offers a comprehensive analytical perspective. By delving into AMR-associated omics data via ML, researchers are paving the way for future research that might unveil intricate biomolecular interactions, propelling computer-aided drug design for life-saving therapeutic treatments.

## 5. Machine Learning for Antibiotic Drug Discovery

ML is emerging as a transformative tool in the realm of antibiotic drug discovery. Since the advent of penicillin, antibiotics have been pivotal in modern medicine. However, the rise of antibiotic-resistant strains and the decline in new antibiotic development pose significant threats to global health. Traditional methods of antibiotic discovery, such as screening soil-dwelling microbes, have been challenged by issues such as the dereplication problem, where the same molecules are repeatedly identified [66]. Furthermore, high-throughput screening, which was once seen as a promising avenue, has not yielded new clinical antibiotics since its inception in the 1980s [67]. The slow pace of drug development and AMR’s persistent threat underscore the urgent need for innovative therapeutic solutions. Recent advances in computational methods, particularly computer-aided drug design (CADD), offer a more efficient approach to drug discovery. CADD techniques, which encompass both structure-based and ligand-based drug design, leverage vast chemical databases and computational models to expedite the drug-development process. The integration of machine learning and artificial intelligence into these computational methods is poised to revolutionize the antibiotic discovery landscape, enabling researchers to explore vast chemical spaces and identify novel antibiotic candidates more efficiently and more cost-effectively. Since traditional methods of drug discovery, especially antibiotics, can be time-consuming and expensive, ML offers a route toward accelerated drug discovery, given its ability to analyze vast amounts of data rapidly and its powerful predictive ability. The marriage of ML and drug discovery shows great promise, considering how well ML algorithms can analyze vast datasets, identify abstract patterns, and predict potential therapeutic compounds, thereby accelerating the research process and enhancing the precision of drug development.

Several databases are available for antibacterial drug design, with ChEMBL [68] being the most comprehensive for small molecules. Other notable databases include CO-ADD [69], SPARK, and the antimicrobial index. ML has been applied to design antibacterial small molecules with impressive results. Yang et al. utilized machine learning to design antibacterial small molecules, achieving prediction accuracies of up to 98.15%. Other studies employed machine learning to predict antibacterial activity and permeation in Gram-negative bacteria, and to design “hybrid” molecules from multiple fragments. ML has been instrumental in targeting mycobacterial infections. Notably, the MycoCSM method, a graph-based decision-tree model, has been used to predict bioactivity against the *Mycobacterium* genus [70]. ML has emerged as a potent tool in bioactivity prediction; enhancing the accuracy of high-throughput virtual screening, employing various approaches such as ligand-based, structure-based, and consensus-based methods. The increasing availability of quality data, coupled with curated and resistance-focused libraries, has further enhanced the effectiveness of machine learning in this domain. Stokes et al. already employed a directed message passing neural network, a type of graph CNN, and identified a new antibiotic, halicin, along with several other potential antibiotic candidates. Halicin has been experimentally validated to be effective against *Staphylococcus aureus* biofilms in vitro [71]. While the potential of machine learning in antibacterial drug design is evident, the field is largely still in its nascent stages. It should be noted that many ML-based drug-design studies are proof-of-concept works, with models primarily tested on data without subsequent experimental biological evaluation. While the insights gained from these studies—especially regarding featurization approaches and methods—are invaluable, in silico predictions alone are insufficient to develop novel chemotherapeutic therapies.

Drawing from an extensive dataset encompassing 63,410 metagenomes and 87,920 microbial genomes, researchers have developed AMPSphere, a comprehensive catalog that houses 863,498 non-redundant peptides [72]. Intriguingly, a significant majority of these peptides were previously undiscovered. Focusing on human-associated microbiota, their study revealed discernible differences at the strain level in AMP production. To substantiate their computational AMP predictions, the researchers chemically synthesized 50 peptide sequences and subjected them to experimental testing against 11 clinically pertinent drug-resistant pathogens; including *Acinetobacter baumannii*, *Escherichia coli* (including one colistin-resistant strain), *Klebsiella pneumoniae*, *Pseudomonas aeruginosa*, *Staphylococcus aureus* (including one methicillin-resistant strain), vancomycin-resistant *Enterococcus faecalis*, and vancomycin-resistant *Enterococcus faecium*. Upon initial screening, 27 AMPs were found to completely inhibit the growth of at least one of the aforementioned pathogens. Further, 72% (32 of 50) of the synthesized AMPs demonstrated antimicrobial activity against commensal or pathogenic strains. Interestingly, some trials yielded bacterial inhibition with AMP concentrations as low as 1 μmol·L^−1^; analogous with MICs discovered among known potent peptides [73]. A standout observation was that the majority of the identified AMPs bore no significant resemblance to existing sequences, underscoring their novelty.

## 6. Concluding Remarks and Future Directions

The potential of ML in revolutionizing various domains, including healthcare, is undeniably profound. However, the successful integration of ML into healthcare practices necessitates a meticulous examination of its inherent limitations and challenges. Drawing from the existing body of literature, we identified drawbacks that must be systematically addressed to harness the full potential of ML in a meaningful and effective manner. These limitations encompass aspects such as data quality, the risk of overfitting, model selection, computational resources, interpretability, continuous updating, and potential bias in training data. The thoughtful consideration and mitigation of these challenges are paramount in the responsible and impactful application of ML in healthcare and beyond. 

For an ML workflow to efficiently discern the genetic features that are instrumental in driving AMR, a model’s success is contingent upon the availability and integrity of high-quality data. Such data are indispensable for the accurate characterization and understanding of the underlying phenomena. Despite the prevalence of data, significant limitations persist within the clinical domain. Even as the European Committee on Antimicrobial Susceptibility Testing (EUCAST) and the Clinical and Laboratory Standards Institute (CLSI) continually update and publish standards on an annual basis, interpretations pertaining to the susceptibility of key antibiotics to common pathogens, such as *Acinetobacter* spp. and *Stenotrophomonas maltophilia*, remain conspicuously absent (Gajic et al., 2022). It is imperative that the provision of laboratory guidelines for all nations persist in reporting epidemiological specificities. This meticulous approach is vital in the fight against AMR, as it fosters the integration of biomedical research and ML for practical application in clinical settings.

Moreover, the expansion of models to incorporate higher dimensionality of data in their architecture, such as 3D structural data, SNPs, and variants, is necessary to further improve our understanding of the complex biological mechanisms that underlie AMR. The success of ML performance is contingent on available, high-quality data in large quantities. While strides have been made in curating robust AMR-related datasets, the absence of standardized repositories and a unified ontology across databases has created barriers to effective data-sharing and collaboration among various institutions, healthcare providers, and governmental entities. AI, though still in the nascent stages of its evolution, is manifesting as an indomitable force in the transformation of healthcare. As methodologies for data collection continue to advance and expand in scope, there is an imperative to align computational strategies and techniques with the breakthroughs that are occurring in biomedical research. This alignment is vital for the development of innovative therapeutic interventions and the enhancement of global health and human well-being.

## Figures and Tables

**Figure 1 antibiotics-12-01604-f001:**
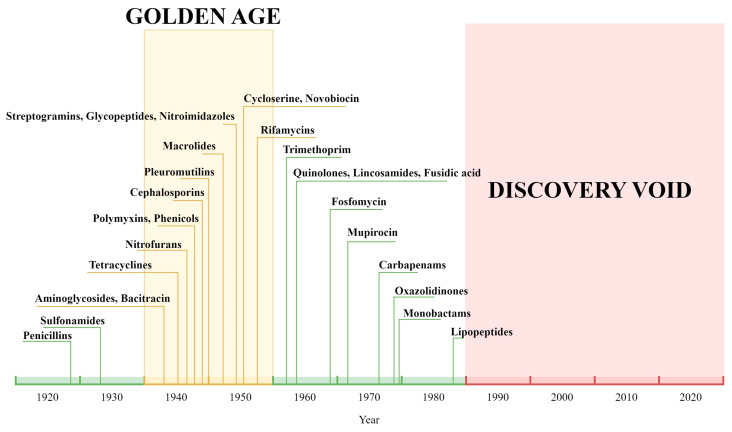
Timeline showcasing the discovery of various antibiotic classes used clinically. The “golden age” spans the period from the 1940s to 1960, during which sources of half of today’s commonly prescribed drugs were discovered. In contrast, the “discovery void” encompasses the period from roughly 1990 to the present, a time in which limited new clinical discoveries have been made [1,3].

**Figure 2 antibiotics-12-01604-f002:**
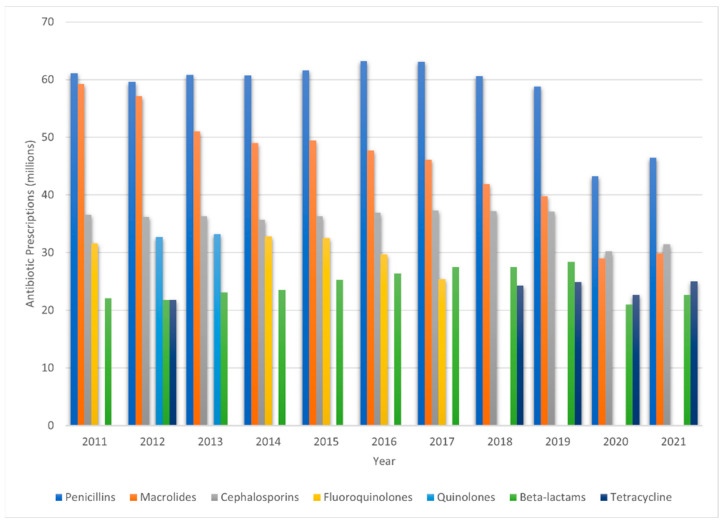
A bar graph showcasing the leading five antibiotic classes and agents prescribed in the United States from 2011 to 2021, measured in millions. The colors differentiate the antibiotic classes, including penicillin, macrolides, cephalosporins, fluoroquinolones, quinolones, beta-lactams, and tetracycline [12].

**Figure 3 antibiotics-12-01604-f003:**
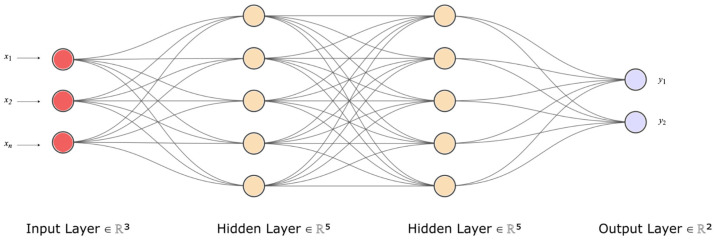
A simplified neural network showcasing the flow of data. The input layer takes in data (*x*_1_, *x*_2_, …, *x*_*n*_), such as sequence information. The hidden layers process and learn relevant features underlying the data, interpreting the data through weighted connections. Finally, the output layer generates outcomes, that is, class prediction (*y*_1_*, y*_2_), such as the binary classification of susceptible and resistant phenotypes. The network adjusts its weights during training to increase the prediction performance, based on the input features and the desired outcomes.

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
