# Peer review of "Silicon versus Superbug: Assessing Machine Learning’s Role in the Fight against Antimicrobial Resistance"

_antibiotics, 2023, doi:10.3390/antibiotics12111604_

Round 1

Reviewer 1 Report

Comments and Suggestions for Authors

This article Assess Machine Learning’s Role in the Fight Against Antimicrobial Resistance. I have some minor suggestions regarding manuscript.

1)                What are the necessary precautions to combat anti microbial resistance?

2)                What is meant by machine learning methodologies?

3)                What are the applications of machine learning in understanding AMR?

4)                What are the challenges in successful implementation of machine learning?

5)                How antibiotics ability to inhibit bacterial infection is assessed?

6)                What are the limitations of antibiotic susceptibility testing?

7)                How antibiotic drug discovery and machine learning are related?

8)                What are the limitations of machine learning?

9)                How does resistance determinants give rise to superbugs?

10)           What is artificial neural network?

11)           What are the categories of machine learning methodology?

Reviewer 2 Report

Comments and Suggestions for Authors

This reviewer reads this review article with interest. I am not an expert in the area of ML, but am convinced that ML will have a great impact on medical biology and the fight against AMR, as indicated by the authors.

When using the keywords: "machine learning" and "antimicrobial resistance" to search for related publications under Pubmed, I see 53 review articles in the past five years. In 2023 only, there were 75 hits when using these keywords to search for publications under PubMed. Therefore, the authors are encouraged to present the most updated knowledge in this review article, avoiding the repetitive description of similar knowledge in your review article as in many other similar articles.

There is a big section in this review article, namely “Limitations of Antibiotic Susceptibility Testing”. This reviewer is not sure of the rationale of such a section in this review article. The inclusion of such a section does not flow well for this reviewer. You brought up the question regarding the limitations of conventional AST. Then, you plan to use ML to address the challenges and limitations.

Just a thought for the authors to consider: It would be interesting if the authors could come up with an eye-catching figure (figures), illustrating the main points in your review article. By doing this, you may separate your review article from many other review articles.

Round 2

Reviewer 2 Report

Comments and Suggestions for Authors

The authors have done a wonderful job in addressing the comments from this review. A great job!